# Human alignment of neural network representations

**Lukas Muttenthaler**[*]
Machine Learning Group
Technische Universität Berlin
BIFOLD[†]

**Lorenz Linhardt**
Machine Learning Group
Technische Universität Berlin
BIFOLD[†]

**Jonas Dippel**
Machine Learning Group
Technische Universität Berlin
BIFOLD[†]

**Robert A. Vandermeulen**
Machine Learning Group
Technische Universität Berlin
BIFOLD[†]

**Simon Kornblith**
Google Brain, Toronto

## Abstract

Today's computer vision models achieve human or near-human level performance across a wide variety of vision tasks. However, their architectures, data, and learning algorithms differ in numerous ways from those that give rise to human vision. In this paper, we investigate the factors that affect alignment between the representations learned by neural networks and human concept representations. Human representations are inferred from behavioral responses in an odd-one-out triplet task, where humans were presented with three images and had to select the odd-one-out. We find that model scale and architecture have essentially no effect on alignment with human behavioral responses, whereas the training dataset and objective function have a much larger impact. Using a sparse Bayesian model of human conceptual representations, we partition triplets by the concept that distinguishes the two similar images from the odd-one-out, finding that some concepts such as food and animals are well-represented in neural network representations whereas others such as royal or sports-related objects are not. Overall, although models trained on larger, more diverse datasets achieve better alignment with humans than models trained on ImageNet alone, our results indicate that scaling alone is unlikely to be sufficient to train neural networks with conceptual representations that match those used by humans.

## 1 Introduction

Representation learning is a fundamental part of modern computer vision systems, but the paradigm has its roots in cognitive science. When Rumelhart et al. [57] developed backpropagation, their goal was to find a method that could learn representations of concepts that are distributed across neurons, similarly to the human brain. The discovery that representations learned by backpropagation could replicate nontrivial aspects of human concept learning was a key factor in its rise to popularity in the late 1980s [65, 45]. A string of empirical successes has since shifted the primary focus of representation learning research away from its similarities to human cognition and toward practical applications. This shift has been fruitful. By some metrics, the best computer vision models now outperform the best individual humans on benchmarks such as ImageNet [60, 8, 69]. However, the extent to which the conceptual representations learned by these high-performing vision models align with those used by humans remains unclear.

---

[*]Also affiliated with the Max Planck Institute for Human Cognitive and Brain Sciences, Leipzig, Germany.
[†]BIFOLD, Berlin Institute for the Foundations of Learning and Data, Berlin, Germany

4th Workshop on Shared Visual Representations in Human and Machine Visual Intelligence (SVRHM) at the Neural Information Processing Systems (NeurIPS) conference 2022. New Orleans.

Do models that are better at classifying images naturally learn more human-like conceptual representations? Prior work has investigated this question indirectly, by measuring models' error consistency with humans [18, 52, 21] and the ability of their representations to predict neural activity in primate brains [71, 23, 59], with mixed results. Here, we approach the question of alignment between human and machine representation spaces more directly, using human similarity judgments collected from an odd-one-out task, where humans saw triplets of images and selected the image most different from the other two [28]. These similarity judgments allow us to infer that the two images that were not selected are closer to each other in an individual's concept space than either is to the odd-one-out. We define the odd-one-out in the neural network representation space analogously, and measure neural networks' alignment with human similarity judgments in terms of their *odd-one-out accuracy*, i.e., the accuracy of their odd-one-out "judgments" with respect to humans', under a wide variety of settings. Based on these odd-one-out accuracies, we draw the following conclusions:

- Scaling ImageNet models improves ImageNet accuracy, but does not consistently improve alignment of their representations with human similarity judgments. Differences in alignment across ImageNet models appear to arise primarily from differences in objective functions rather than from differences in architecture or width/depth.
- Models trained on image/text data, or on larger, more diverse classification datasets than ImageNet, achieve substantially better alignment with humans.
- We use a sparse Bayesian model of human mental representations [44] to partition triplets by the concept that distinguishes the odd-one-out. While food and animal-related concepts can easily be recovered from neural net representations, human alignment is weak for dimensions that depict sports-related or royal objects, especially for ImageNet models.

We discuss related work more thoroughly in Appendix A.

## 2  Methods

**Data** The images used in this paper are taken from the THINGS database [27]. THINGS consists of a collection of 1,854 object categories, i.e., concrete nameable nouns in the English language, along with representative images for these categories. THINGS was curated to include categories that can be easily identified as a central object in a natural image. Hebart et al. [28] collected similarity judgments from human participants on categories in THINGS, which they then used to derive concept repre-

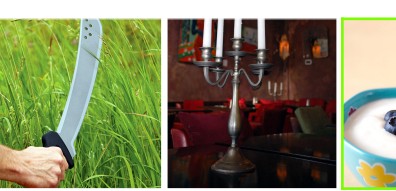

Figure 1: An example triplet from Hebart et al. [28], where neural nets choose a different odd-one-out than a human. The images in this triplet are copyright-free images from THINGS + [64].

sentations. These similarity judgments came in the form of responses to a *triplet task*. In a triplet task, images from three distinct categories are presented to a participant, from which the participant selects the image that is most different from the other two (or equivalently the pair of images that are most similar). The authors collected 1.46 million unique responses crowdsourced from 5,301 workers. See Figure 1 for an example triplet. For presentation purposes, we have replaced the images used in Hebart et al. [28] with images similar in appearance that are licensed under CC0 [64].

**Models** In our evaluation, we consider a diverse set of pretrained neural networks, including a wide variety of self-supervised and supervised models trained on ImageNet-1K and ImageNet-21K [13]; a Vision Transformer trained on the proprietary JFT-3B dataset [73]; and models that were trained on both image and text data such as CLIP [51], ALIGN [32], and BASIC [49]. See Table C.1 for a comprehensive list of all models. In our plots, we determine the ImageNet top-1 accuracy for networks not trained on ImageNet-1K by training a linear classifier on the network's penultimate layer using L-BFGS [41].

**Zero-shot odd-one-out accuracy** We examine the extent to which the odd-one-out can be identified directly from the similarities between images in models' representation spaces. Given representations $x_1$, $x_2$, and $x_3$ of the three images that make up the triplet, we first construct a similarity matrix $S \in \mathbb{R}^{3 \times 3}$ where $S_{i,j} := x_i^T x_j / (\|x_i\|_2 \|x_j\|_2)$, the cosine similarity between a pair of representations. We identify the closest pair of images in the triplet as $\arg\max_{i,j>i} S_{i,j}$; the remaining image is the odd-one-out. We define zero-shot odd-one-out accuracy as the proportion of triplets where the odd-one-out identified in this fashion matches the human odd-one-out response. When evaluating

zero-shot odd-one-out accuracy, we report the better of the accuracies obtained from representations of the penultimate embedding layer and logits (if the network has a logits layer). As we show in Figure C.1, representations obtained from earlier network layers perform worse.

**Probing** In cases where a model's zero-shot accuracy is low, decoding the information necessary for downstream tasks may only require a linear transformation. Generally, linear probing yields insights into the information encoded in neural net's representation [61, 2].

To perform linear probing, we formulate the notion of the odd-one-out probabilistically, as in Hebart et al. [28]. Given similarity matrix $\boldsymbol{S}$ and a triplet $\{i, j, k\}$ (here the images are indexed by natural numbers), the likelihood of a particular pair, $\{a, b\} \subset \{i, j, k\}$, being most similar, and thus the remaining image being the odd-one-out, is modeled by the softmax of the image similarities,

$$p(\{a,b\}|\{i,j,k\}, \boldsymbol{S}) \coloneqq \exp(S_{a,b}) / \left( \exp(S_{i,j}) + \exp(S_{i,k}) + \exp(S_{j,k}) \right). \tag{1}$$

We learn the linear transformation that maximizes the log-likelihood of the triplet odd-one-out judgments plus an $\ell_2$ regularization term. Specifically, given triplet responses $(\{a_s, b_s\}, \{i_s, j_s, k_s\})_{s=1}^n$ we find a square matrix $\boldsymbol{W}$ yielding a similarity matrix $S_{ij} = (\boldsymbol{W}\boldsymbol{x}_i)^T(\boldsymbol{W}\boldsymbol{x}_j)$ that optimizes

$$\arg\min_{\boldsymbol{W}} \quad -\frac{1}{n} \sum_{s=1}^n \log p\left(\{a_s, b_s\}|\{i_s, j_s, k_s\}, \boldsymbol{S}\right) + \lambda ||\boldsymbol{W}||_2^2. \tag{2}$$

Here, we determine $\lambda$ via grid-search during $k$-fold cross-validation (CV). To obtain a minimally biased estimate of the odd-one-out accuracy of a linear probe, we partition the $m$ objects into two disjoint sets. Experimental details about the optimization process, $k$-fold CV, and how we partition the objects can be found in Appendix B.1 and in Algorithm 1 respectively.

**VICE** Several of our analyses make use of human concept representations obtained by Variational Interpretable Concept Embeddings (VICE), an approximate Bayesian method for finding concept representations from human odd-one-out responses in a triplet task [44]. VICE uses mean-field VI to yield a sparse representation for each image that best explains these responses. VICE does not consider the content of the images and cannot provide representations for novel images. VICE shows high reproducibility of representations across different random initializations, and has strong predictive power, achieving an odd-one-out accuracy of $\sim 64\%$ on THINGS, which is only marginally lower than the estimated ceiling accuracy of $67.22\%$ [28].

## 3   Experiments

Here, we investigate how closely neural networks' representation spaces align with humans' concept spaces, and whether concepts can be recovered from a representation via a linear transformation.

### 3.1   Odd-one-out vs. ImageNet accuracy

We begin by comparing zero-shot odd-one-out accuracy for THINGS with ImageNet accuracy for all models in Table C.1. ImageNet accuracy generally is a good predictor for transfer learning performance [35, 14, 17]. Thus, we evaluate zero-shot odd-one-out accuracy for all models in Table C.1 and compare it with their ImageNet top-1 accuracy. Figure 2 shows results for both the THINGS triplet task (left) as well as a

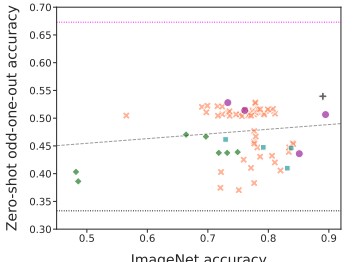 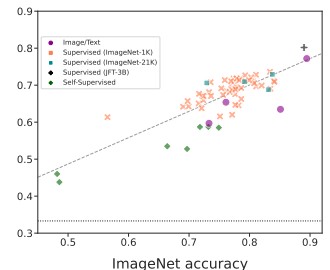

Figure 2: Zero-shot odd-one-out accuracy as a function of ImageNet accuracy for THINGS (*left*) and CIFAR-100 coarse (*right*). Dashed diagonal lines indicate a least-squares fit. Dashed horizontal lines reflect chance-level or ceiling accuracy respectively.

triplet task constructed using the 20 coarse classes of the CIFAR-100 dataset (right). To generate CIFAR-100 triplets, we select two images from the same coarse class and one odd-one-out image from a different class; see Appendix D for further details. While ImageNet accuracy is highly correlated with odd-one-out accuracy for the CIFAR-100 coarse task ($r = 0.809$), its correlation with

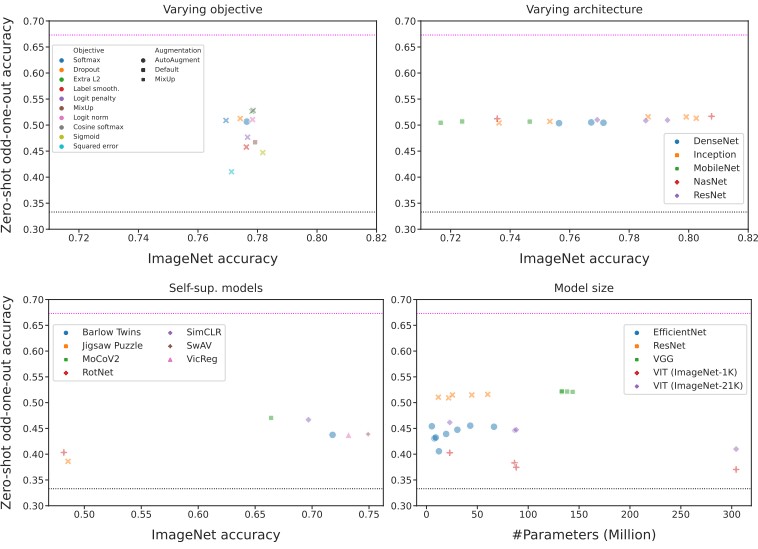

Figure 3: Zero-shot odd-one-out accuracy as a function of ImageNet accuracy or number of model parameters. **Top**: Models on the left have the same architecture (ResNet-50) but were trained with a different objective function or different data augmentation. Models on the right were trained with the same objective function but vary in architecture. **Bottom**: Performance for different SSL models on the left, and a subset of ImageNet models with their number of parameters on the right. Dashed horizontal lines reflect chance-level or ceiling accuracy respectively.

accuracy on human odd-one-out judgments is much weaker ($r = 0.131$). This raises the question whether there are model, task, or data characteristics that influence human alignment.

**Architecture or objective?** The top row of Figure 3 shows odd-one-out accuracy as a function of ImageNet performance for models from two recent studies that investigated the transferability of ImageNet pretrained representations that vary only in the architecture or training objective, with all other hyperparameters fixed [35, 36]. We find that models with the same architecture (ResNet-50) trained with different data augmentation or objective functions [36] yield substantially different zero-shot odd-one-out accuracies. Conversely, models with different architectures trained with the same objective function [35] - softmax cross-entropy -, achieve similar odd-one-out accuracies, although their ImageNet accuracies vary significantly. This suggests that architecture does not affect odd-one-out accuracy, while the objective function and the augmentation strategy have a significant impact.

**Self-supervised learning** The plot in the bottom left corner of Figure 3 compares zero-shot odd-one-out accuracy of different SSL models with their linear probing ImageNet performance. The non-Siamese models Jigsaw [46] and RotNet [22] show substantially worse alignment with human judgments than other SSL models. This is not surprising given their poor performance on ImageNet. For the Siamese methods SimCLR [11], MoCoV2 [26], Barlow Twins [72], SwAV [10], and VICReg [6], however, ImageNet performance does not correspond to alignment with human judgments.

**Model capacity** The graph in the bottom right corner of Figure 3 plots zero-shot odd-one-out accuracy against the number of model parameters for a subset of ImageNet models. While one typically observes a positive correlation between model capacity and task performance in computer vision, we do not observe any relationship between model width/depth and odd-one-out accuracy.

### 3.2 How much alignment can a linear probe recover?

Probing and zero-shot odd-one-out accuracies are positively correlated, in the embedding (Figure 4; $r = 0.645$) and logits layer (Figure E.2; $r = 0.963$). However, there are models in Figure 4 that show poor zero-shot and strong linear probing odd-one-out accuracies, such as ALIGN, SWAV, EfficientNet B4 and ViT-L/16. ALIGN is probably the most interesting candidate. Although its zero-shot odd-one-out accuracy is average, this image/text model achieves the highest probing odd-one-out accuracy across all evaluated models.

As we show in Appendix E, the relationship between probing odd-one-out accuracy and ImageNet accuracy is

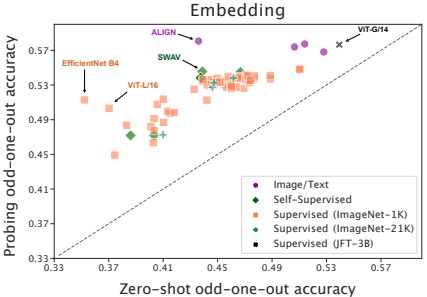

Figure 4: Zero-shot and probing odd-one-out accuracies for the embedding layer of all neural nets. Dashed line indicates $x = y$.

similar to the relationship between zero-shot odd-one-out accuracy and ImageNet accuracy described above. The correlation between ImageNet accuracy and probing odd-one-out accuracy is still weak ($r = 0.213$). Probing reduces the variance in odd-one-out accuracy among networks trained with different loss functions and Siamese self-supervised learning methods, but there is still no clear improvement in odd-one-out accuracy with better-performing architectures or larger model capacities.

## 3.3 Human alignment is concept-specific

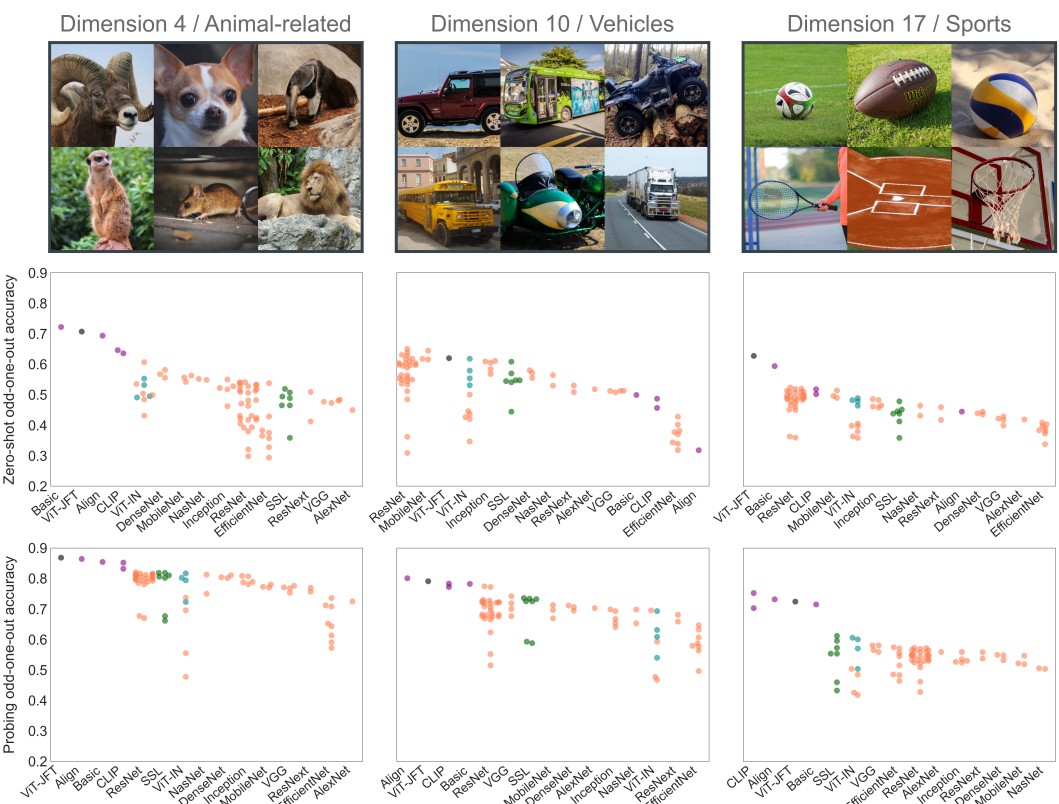

Figure 5: Zero-shot and linear probing odd-one-out accuracies for the embedding layer of all models for a subset of three of the 45 VICE dimensions. Color-coding was determined by training data/objective. Violet: Image/Text. Green: Self-supervised. Orange: Supervised (ImageNet-1K). Cyan: Supervised (ImageNet-21K). **Black**: Supervised (JFT-3B).

In the following analysis, we evaluate both zero-shot and linear probing odd-one-out accuracy for individual human concepts. We partitioned the original triplet dataset according to the VICE dimension shared between the two more similar images; see Appendix F for details. In Figure 5, we show zero-shot and linear probing odd-one-out accuracies for three VICE dimensions, for all models listed in Table C.1.

Although most image/text models and ViT-G/14 JFT showed a higher probing odd-one-out accuracy compared to self-supervised models or models trained on ImageNet, zero-shot odd-one-out accuracy was somewhat less consistent. For dimension 10, ResNets from Kornblith et al. [36], trained with a cosine softmax objective, were the best zero-shot performing models, whereas image/text models' zero-shot performance were among the worst. For dimension 4, an animal-related concept, models pretrained on ImageNet clearly showed the worst performance, whereas this concepts seems to be well represented in image/text models. After linear probing, results became less ambiguous. For almost every human concept, image/text models and ViT-G/14 JFT were the best human aligned models, whereas both AlexNet and EfficientNets achieved the lowest per-concept odd-one-out accuracies. This difference between image/text models and ViT-G/14 JFT and the other ImageNet-prerained models was particularly apparent for dimension 17 which summarizes

sports-related objects. For this dimension, we observed a large performance gap after linear probing between image/text models and ViT-G/14 JFT and all remaining models. Analogously, in Appendix G we perform the same experiments using linear regression to predict representations from VICE. These experiments corroborate the results obtained from linear probing.

## 4 Discussion

In this work, we evaluated the alignment of neural network representation with human concept spaces through performance in an odd-one-out task. Before discussing our findings, we want to address limitations of our work. One obvious limitation is the fact that we did not consider non-linear transformations. It is possible that there exist families of simple non-linear transformations that can provide better alignment for the networks we investigate. We plan to investigate such transformations more thoroughly in future work. Another limitation relates to the use of pretrained models for our experiments. These models have been trained with a variety of objectives and regularization strategies. We have mitigated this limitation by comparing controlled subsets of models in Figure 3.

Nevertheless, we can draw the following conclusions from our findings. First, scaling ImageNet models does not lead to better alignment of their representations with human similarity judgments. Differences in human alignment across ImageNet models are mainly attributable to the objective function with which a model was trained, whereas architecture and model capacity are both insignificant. Second, models trained on image/text or more diverse data achieve much better alignment than ImageNet models. Albeit not consistent for zero-shot odd-one-out accuracy, this is clear in both linear probing and regression results. Third, good representations of concepts that are important to human similarity judgments can be recovered from neural network representation spaces. However, representations of less important concepts, such as `sports` and `royal` objects, are more difficult to recover.

How can we train neural networks that achieve better alignment with human concept spaces? Although our results indicate that large, diverse datasets improve alignment, all image/text and JFT models we investigate all attain probing accuracies of 57-58%. By contrast, VICE representations achieve 64%, and a Bayes-optimal classifier achieves 67%. Since our image/text models are trained on datasets of varying sizes (400M to 6.6B images) but achieve similar alignment, we suspect that further scaling of dataset size is unlikely to close this gap. To obtain substantial improvements, it may be necessary to incorporate additional forms of supervision when training the representation itself. Benefits of improving human/machine alignment may extend beyond accuracy on our triplet task, to transfer and retrieval tasks where it is important to capture human notions of similarity.

**Acknowledgments**

LM, LL, JD and RV acknowledge support by the Federal Ministry of Education and Research (BMBF) for the Berlin Institute for the Foundations of Learning and Data (BIFOLD) (01IS18037A). We thank Robert Geirhos and Andrew Lampinen for their helpful comments on earlier versions of the manuscript.

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

# A   Related Work

Most work comparing neural networks with human behavior has focused on the errors made during image classification. Although ImageNet-trained models appear to make very different errors than humans [52, 20, 21], models trained on larger datasets produce more consistent errors [21], consistent with our findings here. Compared to humans, ImageNet-trained models perform worse on distorted images [54, 15, 31, 18] and rely more heavily on texture cues and less on object shapes [19, 5], although reliance on texture can be mitigated through data augmentation [19, 30, 40], adversarial training [21], or larger datasets [9].

Previous work has also compared human and machine semantic similarity judgments, generally using smaller sets of images and models than we explore here. Jozwik et al. [33] measured the similarity of AlexNet and VGG-16 representations to human similarity judgments of 92 object images inferred from a multi-arrangement task. Peterson et al. [48] compared representations of five neural networks to similarity judgments for six different sets of 120 images, obtained by asking subjects to rate the similarities of pairs from 0 to 10. They report results both with and without rescaling of features. Attarian et al. [4] learned constrained linear transformations of representations to improve the fit of VGG-16 representations to similarity judgments for bird images, but found that unconstrained transformations perform best. Aminoff et al. [3] found that, across 11 networks, representations of contextually associated objects (e.g., bicycles and helmets) were more similar than those of non-associated objects; similarity correlated with both human ratings and reaction times. Roads & Love [55] collect human similarity judgments for the ImageNet validation set and evaluate triplet accuracy on these similarity judgments using 12 ImageNet networks. Most closely related to our work, Marjieh et al. [42] compare similarity of representations of 611 models to cardinal pairwise human similarity judgments. They find a weak correlation between parameter count and models' similarities with humans, and find that incorporating embeddings of both image and text models can further improve the correlation. However, they do not attempt to systematically examine factors that affect alignment between image models and human similarity judgments.

Other studies have focused on perceptual similarity rather than semantic similarity, where the task measures perceived similarity between a reference image and a distorted version of that reference image [50, 74], rather than between distinct images as in our task. Whereas the representations best aligned with human perceptual similarity are obtained from intermediate layers of small architectures [7, 74, 12, 39], the representations best aligned with our odd-one-out judgments are obtained at final model layers, and architecture has little impact.

Our work fits into a broader literature examining relationships between in-distribution accuracy of image classification and other model quality measures, such as accuracy on out-of-distribution

data and downstream accuracy when transferring the model. Out-of-distribution accuracy correlates nearly linearly with accuracy on the training distribution [53, 68, 43], although certain forms of data augmentation can improve accuracy under some distribution shifts without an accompanying improvement in in-distribution accuracy [29]. When comparing the transfer learning performance of different architectures trained with similar settings, accuracy on the pretraining task correlates well with accuracy on the transfer tasks [35], although differences in regularization, training objective, and hyperparameters can have a substantial impact on linear transfer accuracy even if the impact on pretraining accuracy is small [35, 36, 1]. In our study, we find that the training objective has a significant impact, as it does for linear transfer. However, in contrast to previous observations regarding out-of-distribution generalization and transfer, we find that better-performing architectures do not achieve greater human alignment.

## B    Experimental details

### B.1    Linear probing

**Initialization** We initialized the transformation matrix $W \in \mathbb{R}^{p \times p}$ used in Equation 2 with a temperature scaled identity matrix $\tau I \in \mathbb{R}^{p \times p}$ such that $W := \tau I$ at the beginning of the optimization process. $\tau$ is model-specific and was found via grid search, minimizing the expected calibration error (ECE) [24]. Details on temperature scaling are described in B.2.

**Training** We optimized the transformation matrix $W$ via gradient descent, using Adam [34] with a learning rate of $\eta = 0.001$. We performed a grid-search over the learning rate $\eta$, where $\eta \in \{0.0001, 0.001, 0.01\}$ and found $0.001$ to work best for all models in Table C.1. We trained the linear probe for a maximum of 100 epochs and stopped the optimization process early whenever the generalization performance did not change by a factor of $0.0001$ for $T = 10$ epochs.

**Cross-validation** To obtain a minimally biased estimate of the odd-one-out accuracy of a linear probe, we performed $k$-fold CV over objects rather than triplets. We partitioned the $m$ objects into two disjoint sets for train and test triplets. Algorithm 1 demonstrates how object partitioning was performed for each of the $k$ folds.

Note that the number of train objects that is sampled uniformly at random without replacement from the set of all objects is dependent on $k$. We performed a grid-search search over $k$, where $k \in \{2, 3, 4, 5\}$, and observed that 3-fold and 4-fold CV lead to the best linear probing results. Since objects between train and test triplets were not allowed to overlap, loss of data was inevitable (see Algorithm 1). One can easily see that minimizing the loss of triplet data, comes at the cost of disproportionally decreasing the size of the test set. We decided to proceed with 3-fold CV in our final experiments since using $2/3$ of the objects for training and $1/3$ for testing resulted in a proportionally larger test set than using $3/4$ for training and $1/4$ for testing ($\sim 433$k train and $\sim 54$k test triplets for 3-fold CV vs. $\sim 616$k train and $\sim 23$k test triplets for 4-fold CV). In general, the larger a test set, the more accurate the estimate of a model's generalization performance. To find the optimal strength of the $\ell_2$ regularization for each linear probe, we performed a grid-search over $\lambda$ for each $k$ value individually. The optimal $\lambda$ varied between models, where $\lambda \in \{0.0001, 0.001, 0.01, 0.1, 1\}$.

### B.2    Temperature scaling

It is widely known that classifiers trained to minimize cross-entropy tend to be overconfident in their predictions [66, 24, 56], which is in stark contrast to the high-entropy predictions of VICE. We found it helpful to initialize the transformation matrices for the probing experiments using a temperature parameter, as described in Appendix B.1. For this purpose, we performed temperature scaling [24] on the model outputs for THINGS and searched over the scaling parameter $\tau$ for each model. In particular, we considered temperature-scaled predictions

$$p(\{a, b\}|\{i, j, k\}, \tau S) = \frac{\exp(\tau S_{a,b})}{\exp(\tau S_{i,j}) + \exp(\tau S_{i,k}) + \exp(\tau S_{j,k})},$$

where we multiply $S$ in Equation 1 by a constant $\tau > 0$ and $S_{i,j}$ is the inner product of the model representations for images $i$ and $j$, i.e. the zero-shot similarities. There are several conceivable criteria that could be minimized to find the optimal scaling parameter $\tau$ from a set of candidates. For our analyses we considered the following,

**Algorithm 1** Algorithm for object partitioning during $k$-fold CV

---

**Input:** $(\mathcal{D}, m)$        $\triangleright$ Here, $\mathcal{D} := (\{a_s, b_s\}, \{i_s, j_s, k_s\})_{s=1}^n$ and $m$ is the number of objects

   $[m] = \{1, \ldots, m\}$                                                   $\triangleright |[m]| = m$

   $\mathbb{O}_{\text{train}} \sim \mathcal{U}([m])$    $\triangleright$ Sample a number of train objects uniformly at random without replacement

   $\mathbb{O}_{\text{test}} := [m] \setminus \mathbb{O}_{\text{train}}$                                   $\triangleright$ Test objects are the remaining objects

   $\mathcal{D}_{\text{train}} := \{\}$                                   $\triangleright$ Initialize an empty set for the train triplets

   $\mathcal{D}_{\text{test}} := \{\}$                                     $\triangleright$ Initialize an empty set for the test triplets

   **for** $s \in \{1, \ldots, n\}$ **do**

       assignments $\triangleq$ list( )    $\triangleright$ For each triplet initialize an empty list to control object assignments

       **for** $x \in \{i_s, j_s, k_s\}$ **do**

          **if** $(x \in \mathbb{O}_{\text{train}})$ **then**

             assignment $\triangleq$ "train"

          **else**

             assignment $\triangleq$ "test"

          **end if**

          assignments $\leftarrow$ assignment        $\triangleright$ Append current assignment to the list of assignments

       **end for**

       **if** $(\text{len}(\text{set}(\text{assignments})) \neq 1)$ **then**

          **continue**    $\triangleright$ If not all objects in a triplet belong to the same set of objects, discard triplet

       **else**

          assignment $\triangleq$ pop(set(assignments))        $\triangleright$ Get object set assignment of current triplet

          **if** (assignment **is** "train") **then**

             $\mathcal{D}_{\text{train}} := \mathcal{D}_{\text{train}} \cup \mathcal{D}_s$                  $\triangleright$ Assign current triplet to the train set

          **else**

             $\mathcal{D}_{\text{test}} := \mathcal{D}_{\text{test}} \cup \mathcal{D}_s$                   $\triangleright$ Assign current triplet to the test set

          **end if**

       **end if**

   **end for**

**Output:** $(\mathcal{D}_{\text{train}}, \mathcal{D}_{\text{test}})$                          $\triangleright$ Return both train and test triplet sets

---

- Average Jensen-Shannon (JS) distance between model zero-shot probabilities and VICE probabilities over all triplets
- Average Kullback-Leibler divergence (KLD) between model zero-shot probabilities and VICE probabilities over all triplets
- Expected Calibration Error (ECE) [24].

The ECE is defined as follows. Let $\mathcal{D} = (\{a_s, b_s\}, \{i_s, j_s, k_s\})_{s=1}^n$ be the set of triplets and human responses from Hebart et al. [28]. For a given triplet $\{i, j, k\}$ and similarity matrix $\boldsymbol{S}$ we define confidence as

$$\text{conf}(\{i, j, k\}, \boldsymbol{S}) := \max_{\{a,b\} \subset \{i,j,k\}} p(\{a, b\} \mid \{i, j, k\}, \boldsymbol{S}).$$

This corresponds to the expected accuracy of the Bayes classifier for that triplet according to the probability model from $\boldsymbol{S}$ with Equation 1. We define $B_m(\boldsymbol{S})$ to be those training triplets where

$$\text{conf}(\{i_s, j_s, k_s\}, \boldsymbol{S}) \in \left[\frac{m-1}{10}, \frac{m}{10}\right].$$

For a similarity matrix, $\boldsymbol{S}$, and a set of triplets with responses, $\mathcal{D}' \subset \mathcal{D}$, we define $\text{acc}(\mathcal{D}', \boldsymbol{S})$ to be the portion of triplets in $\mathcal{D}'$ correctly classified according to the highest similarity according to $\boldsymbol{S}$. Finally for a set of triplets $\mathcal{D}' \subset \mathcal{D}$ and similarity matrix $\boldsymbol{S}$ we define $\text{conf}(\mathcal{D}')$ to be the average confidence over that set (triplet responses are simply ignored). The ECE is defined as

$$\text{ECE}(\tau, \boldsymbol{S}) = \sum_{m=1}^{10} \frac{|B_m(\tau\boldsymbol{S})|}{n} |\text{acc}(B_m(\tau\boldsymbol{S})) - \text{conf}(B_m(\tau\boldsymbol{S}))|.$$

Intuitively, the ECE is low if for each subset $B_m(\tau\boldsymbol{S})$ the model's accuracy and its confidence are near each other. A model will be well-calibrated if its confidence in predicting the odd-one-out in a triplet corresponds to the probability that this prediction is correct.

Of the three considered criteria, ECE resulted in the clearest optima when varying $\tau$, whereas KLD plateaued with increasing $\tau$ and JS distance was numerically unstable, most probably because the model output probabilities were near zero for some pairs, which may result in very large JS distance. For all models, we performed a grid-search over $\tau \in \{1 \cdot 10^0, 7.5 \cdot 10^{-1}, 5 \cdot 10^{-1}, 2.5 \cdot 10^{-1}, 1 \cdot 10^{-1}, 7.5 \cdot 10^{-2}, 5 \cdot 10^{-2}, 2.5 \cdot 10^{-2}, 1 \cdot 10^{-2}, 7.5 \cdot 10^{-3}, 5 \cdot 10^{-3}, 2.5 \cdot 10^{-3}, 1 \cdot 10^{-3}, 5 \cdot 10^{-4}, 1 \cdot 10^{-4}, 5 \cdot 10^{-5}, 1 \cdot 10^{-5}\}$.

### B.3 Linear regression

**Cross-validation** We used ridge regression, that is $\ell_2$-regularized linear regression, to find the transformation matrix $\boldsymbol{A}_{j,:}$ and bias $b_j$ that result in the best fit. We employed nested $k$-fold CV for each of the $d$ VICE dimensions. For the outer CV we performed a grid-search over $k$, where $k \in \{2, 3, 4, 5\}$, similarly to how $k$-fold CV was performed for linear probing (see B.1). For our final experiments, we used 5-fold CV to obtain a minimally biased estimate for the $R^2$ score of the regression fit. For the inner CV, we leveraged leave-one-out CV to determine the optimal $\alpha$ for Equation 3 using `RidgeCV` from Pedregosa et al. [47]. We performed a grid search over $\alpha$, where $\alpha \in \{0.01, 0.1, 1, 10, 100, 1000, 10000, 100000, 1000000\}$.

## C  Models

First, we evaluate supervised models trained on ImageNet [58], such as AlexNet [37], various VGGs [62], ResNets [25], EfficientNets [67], ResNext models [70], and Vision Transformers (ViTs) trained on ImageNet-1K[16] or ImageNet-21K [63] respectively. Second, we analyze recent state-of-the-art models trained on image/text data, CLIP-RN & CLIP-ViT [51], ALIGN [32] and BASIC-L [49]. Third, we evaluate self-supervised (SSL) models that were trained with a contrastive learning objective such as SimCLR [11] and MoCo [26], recent SSL models that were trained with a non-contrastive learning objective

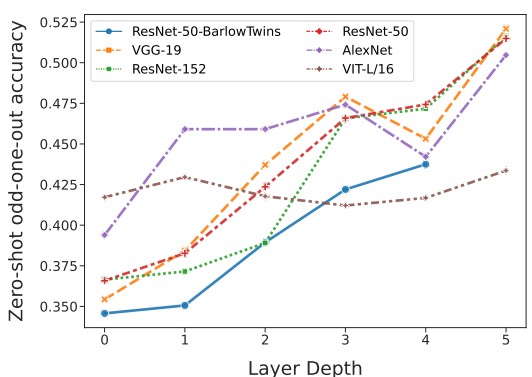

Figure C.1: Zero-shot odd-one-out accuracy at different layers for a subset of selected models.

(no negative examples), BarlowTwins [72], SwAV [10], and VICReg [6], as well as earlier SSL, non-Siamese models, Jigsaw [46], and Rotnet [22]. Last, we evaluate the largest available ViT [73], trained on the proprietary JFT-3B image classification dataset, which consists of approximately three billion images belonging to approximately 30,000 classes [73]. See Table C.1 for further details regarding the models used. Figure C.1 shows the odd-one-out accuracy as a function of layer depth in a neural network for a few different network architectures. Later layers generally perform better which is why we performed our analyses exclusively for the logits or penultimate/embedding layers of the models in Table C.1.

## D  CIFAR-100 triplet task

In a similar vein to the THINGS triplet task, we constructed a reference triplet task from the CIFAR-100 dataset [38]. To show pairs of images that are similar to each other, but do not depict the same object, we leverage the 20 coarse classes of the dataset rather than the original fine-grained classes. For each triplet, we sample two images from the same and an one odd-one-out image from a different coarse class. We

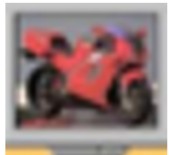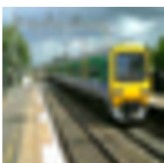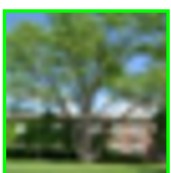

Figure D.1: An example triplet from the CIFAR-100 coarse dataset. The left two images are from one of the two CIFAR-100 "vehicle" superclasses, so the rightmost image is the odd-one-out.

| Model | Source | Architecture | Dataset | Objective | ImageNet Acc. |
|---|---|---|---|---|---|
| AlexNet | [37] | AlexNet | ImageNet-1K | Supervised (softmax) | 56.52% |
| ALIGN | [32] | EfficientNet | ALIGN dataset | Image/Text (contrastive) | 85.11% |
| Basic-L | [49] | CoAtNet | ALIGN + JFT-5B | Image/Text (contrastive) | 89.45% |
| CLIP ResNet-50 | [51] | ResNet | CLIP dataset | Image/Text (contrastive) | 73.30% |
| CLIP ViT-B/32 | [51] | ViT | CLIP dataset | Image/Text (contrastive) | 76.10% |
| DenseNet-121 | [35] | DenseNet | ImageNet-1K | Supervised (softmax) | 75.64% |
| DenseNet-169 | [35] | DenseNet | ImageNet-1K | Supervised (softmax) | 76.73% |
| DenseNet-201 | [35] | DenseNet | ImageNet-1K | Supervised (softmax) | 77.14% |
| EfficientNet B0 | [67] | EfficientNet | ImageNet-1K | Supervised (softmax) | 77.69% |
| EfficientNet B1 | [67] | EfficientNet | ImageNet-1K | Supervised (softmax) | 78.64% |
| EfficientNet B2 | [67] | EfficientNet | ImageNet-1K | Supervised (softmax) | 80.61% |
| EfficientNet B3 | [67] | EfficientNet | ImageNet-1K | Supervised (softmax) | 82.01% |
| EfficientNet B4 | [67] | EfficientNet | ImageNet-1K | Supervised (softmax) | 83.38% |
| EfficientNet B5 | [67] | EfficientNet | ImageNet-1K | Supervised (softmax) | 83.44% |
| EfficientNet B6 | [67] | EfficientNet | ImageNet-1K | Supervised (softmax) | 84.01% |
| EfficientNet B7 | [67] | EfficientNet | ImageNet-1K | Supervised (softmax) | 84.12% |
| Inception-ResNet V2 | [35] | Inception | ImageNet-1K | Supervised (softmax) | 80.26% |
| Inception-V1 | [35] | Inception | ImageNet-1K | Supervised (softmax) | 73.63% |
| Inception-V2 | [35] | Inception | ImageNet-1K | Supervised (softmax) | 75.34% |
| Inception-V3 | [35] | Inception | ImageNet-1K | Supervised (softmax) | 78.64% |
| Inception-V4 | [35] | Inception | ImageNet-1K | Supervised (softmax) | 79.92% |
| MobileNet-V1 | [35] | MobileNet | ImageNet-1K | Supervised (softmax) | 72.39% |
| MobileNet-V2 | [35] | MobileNet | ImageNet-1K | Supervised (softmax) | 71.67% |
| MobileNet-V2 (1.4) | [35] | MobileNet | ImageNet-1K | Supervised (softmax) | 74.66% |
| NASNet-L | [35] | NASNet | ImageNet-1K | Supervised (softmax) | 80.77% |
| NASNet-Mobile | [35] | NASNet | ImageNet-1K | Supervised (softmax) | 73.57% |
| ResNet-50-BarlowTwins | [72] | ResNet | ImageNet-1K | Self-sup. (non-contrastive) | 71.80% |
| ResNet-50-Jigsaw | [46] | ResNet | ImageNet-1K | Self-sup. (non-Siamese) | 48.57% |
| ResNet-50-MoCo-v2 | [26] | ResNet | ImageNet-1K | Self-sup. (contrastive) | 66.40% |
| ResNet-50-RotNet | [22] | ResNet | ImageNet-1K | Self-sup. (non-Siamese) | 48.20% |
| ResNet-50-SimCLR | [11] | ResNet | ImageNet-1K | Self-sup. (contrastive) | 69.68% |
| ResNet-50-SWAV | [10] | ResNet | ImageNet-1K | Self-sup. (non-contrastive) | 74.92% |
| ResNet-50-VICReg | [6] | ResNet | ImageNet-1K | Self-sup. (non-contrastive) | 73.20% |
| ResNet-18 | [25] | ResNet | ImageNet-1K | Supervised (softmax) | 69.76% |
| ResNet-34 | [25] | ResNet | ImageNet-1K | Supervised (softmax) | 73.31% |
| ResNet-50 | [25] | ResNet | ImageNet-1K | Supervised (softmax) | 76.13% |
| ResNet-101 | [25] | ResNet | ImageNet-1K | Supervised (softmax) | 77.37% |
| ResNet-152 | [25] | ResNet | ImageNet-1K | Supervised (softmax) | 78.31% |
| ResNet-101 | [35] | ResNet | ImageNet-1K | Supervised (softmax) | 78.56% |
| ResNet-152 | [35] | ResNet | ImageNet-1K | Supervised (softmax) | 79.29% |
| ResNet-50 | [35] | ResNet | ImageNet-1K | Supervised (softmax) | 76.93% |
| ResNet-50 | [36] | ResNet | ImageNet-1K | Supervised (softmax) | 77.42% |
| ResNet-50 (extra weight decay) | [36] | ResNet | ImageNet-1K | Supervised (softmax+) | 77.82% |
| ResNet-50 (label smoothing) | [36] | ResNet | ImageNet-1K | Supervised (softmax+) | 77.63% |
| ResNet-50 (logit penality) | [36] | ResNet | ImageNet-1K | Supervised (softmax+) | 77.67% |
| ResNet-50 (mixup) | [36] | ResNet | ImageNet-1K | Supervised (softmax+) | 77.92% |
| ResNet-50 (AutoAugment) | [36] | ResNet | ImageNet-1K | Supervised (softmax) | 77.64% |
| ResNet-50 (logit norm) | [36] | ResNet | ImageNet-1K | Supervised (softmax+) | 77.83% |
| ResNet-50 (cosine softmax) | [36] | ResNet | ImageNet-1K | Supervised (softmax+) | 77.86% |
| ResNet-50 (sigmoid) | [36] | ResNet | ImageNet-1K | Supervised (sigmoid) | 78.18% |
| ResNet-50 (softmax) | [36] | ResNet | ImageNet-1K | Supervised (softmax) | 76.94% |
| ResNet-50 (squared error) | [36] | ResNet | ImageNet-1K | Supervised (squared error) | 77.13% |
| ResNeXt-101 32x8d | [70] | ResNeXt | ImageNet-1K | Supervised (softmax) | 79.32% |
| ResNeXt-50 32x4d | [70] | ResNeXt | ImageNet-1K | Supervised (softmax) | 81.11% |
| VGG-11 | [62] | VGG | ImageNet-1K | Supervised (softmax) | 69.02% |
| VGG-13 | [62] | VGG | ImageNet-1K | Supervised (softmax) | 69.93% |
| VGG-16 | [62] | VGG | ImageNet-1K | Supervised (softmax) | 71.59% |
| VGG-19 | [62] | VGG | ImageNet-1K | Supervised (softmax) | 72.38% |
| ViT-B/16 I1K | [63] | ViT | ImageNet-1K | Supervised (sigmoid) | 77.66% |
| ViT-B/16 I21K | [63] | ViT | ImageNet-21K | Supervised (sigmoid) | 83.77% |
| ViT-B/32 I1K | [63] | ViT | ImageNet-1K | Supervised (sigmoid) | 72.08% |
| ViT-B/32 I21K | [63] | ViT | ImageNet-21K | Supervised (sigmoid) | 79.16% |
| ViT-L/16 I1K | [63] | ViT | ImageNet-1K | Supervised (sigmoid) | 75.11% |
| ViT-L/16 I21K | [63] | ViT | ImageNet-21K | Supervised (sigmoid) | 83.13% |
| ViT-S/32 I1K | [63] | ViT | ImageNet-1K | Supervised (sigmoid) | 72.18% |
| ViT-S/32 I21K | [63] | ViT | ImageNet-21K | Supervised (sigmoid) | 72.93% |
| ViT-G/14 JFT | [73] | ViT | JFT-3B | Supervised (sigmoid) | 89.01% |
| ViT-B-16 | [16] | ViT | ImageNet-1K | Supervised (softmax) | 81.07% |
| ViT-B-32 | [16] | ViT | ImageNet-1K | Supervised (softmax) | 75.91% |

Table C.1: Pretrained neural networks that we considered in our analyses.

restrict ourselves to examples from the CIFAR-
100 train set and exclude the validation set. We randomly sample a total of 50,000 triplets which is equivalent to the size of the original train set. Figure D.1 shows an example triplet for this task.

# E    Linear probing

In the left plot of Figure E.1, we show probing odd-one-out accuracy as a function of ImageNet accuracy for all models in Table C.1. Similarly to the findings depicted in Figure 2, we observe a low Pearson correlation coefficient ($r = 0.241$) between ImageNet accuracy and probing odd-one-out accuracy. As a reference, here we show again ImageNet accuracy as a function of zero-shot odd-one-out accuracy on the CIFAR-100 coarse triplet task. In Figure E.2 we compare probing odd-one-out accuracy with zero-shot odd-one-out accuracy for models pretrained on ImageNet-1K or ImageNet-21K. We observe a strong positive correlation of $r = 0.963$ between probing odd-one-out and zero-shot odd-one-out accuracy.

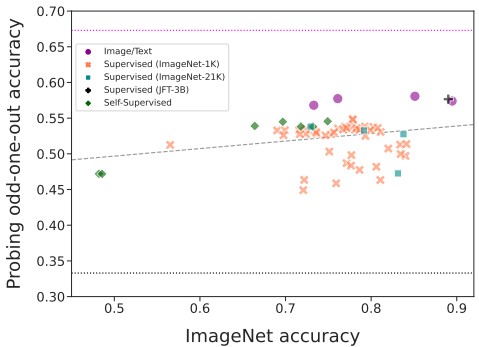

Figure E.1: Probing odd-one-out accuracy as a function of ImageNet accuracy. Dashed diagonal line indicate a least-squares fit. Dashed horizontal lines reflect chance-level or ceiling accuracy respectively.

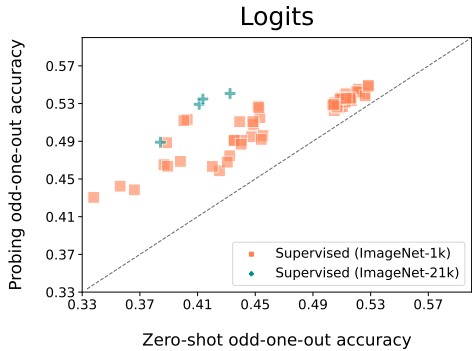

Figure E.2: Probing odd-one-out accuracy as a function of zero-shot odd-one-out accuracy for the logits layer of all ImageNet models in Table C.1. Dashed line indicates $x = y$ line.

# F    Human alignment is concept specific

To examine how well neural nets represent human concepts, we partitioned the original triplet dataset $\mathcal{D}$ into two sets $\mathcal{D}^*$ and $\mathcal{D}^\dagger$, with $\mathcal{D}^*$ containing triplets correctly predicted by VICE and $\mathcal{D}^\dagger$ containing those which are not. The triplets in $\mathcal{D}^\dagger$ mostly have high entropy, i.e., chosen odd-one-out is not consistent for humans. The triplets in $\mathcal{D}^\dagger$ are not used in the following analysis. We further partitioned $\mathcal{D}^*$ into 45 subsets according to the 45 VICE dimensions, $\mathcal{D}_1^*, \ldots, \mathcal{D}_{45}^*$. A triplet belongs to $\mathcal{D}_j^*$ when the sum of the VICE representations for the two most similar objects in the triplet, $\boldsymbol{x}_a$, $\boldsymbol{x}_b$, attains its maximum in dimension $j$, i.e. $j = \arg\max_{j'} x_{a,j'} + x_{b,j'}$.

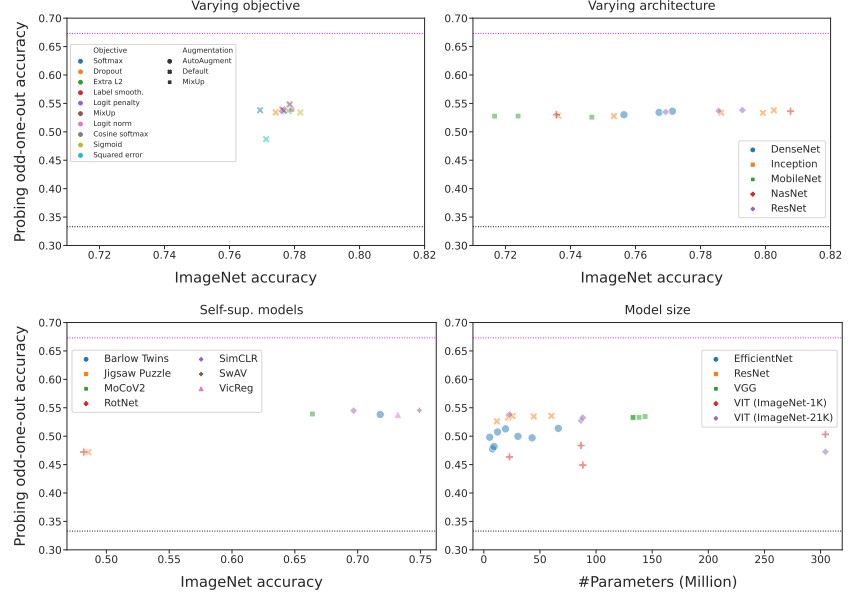

Figure E.3: Probing odd-one-out accuracy for THINGS as a function of ImageNet accuracy or number of model parameters. **Top**: Models on the left have the same architecture (ResNet-50) but were trained with a different objective function [36]. Models on the right were trained with the same objective function but vary in architecture [35]. **Bottom**: Performance for different SSL models on the left, and a subset of ImageNet models with their number of parameters on the right. Dashed horizontal lines reflect chance-level or ceiling accuracy respectively.

# G    Linear regression

## G.1    Overall performance

In Figure G.1, we compare odd-one-out accuracies after linear probing with zero-shot odd-one-out accuracies and probing odd-one-out accuracies for logits vs. embedding layers of ImageNet models. The results are consistent with the results from linear probing shown in Figure 4.

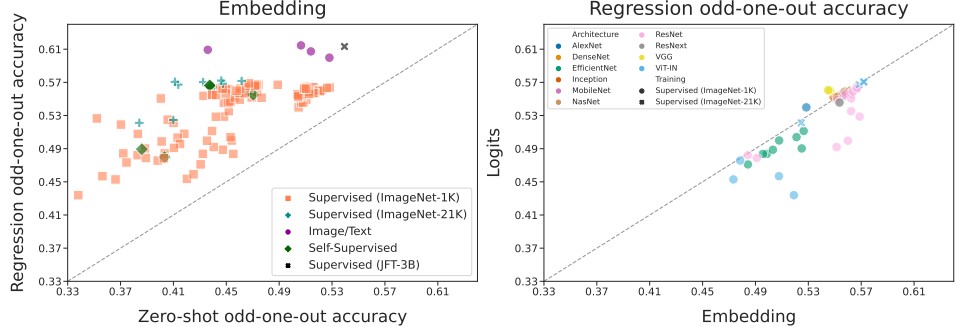

Figure G.1: **Left**: Zero-shot and regression odd-one-out accuracies for the embedding layer of all neural nets. **Right**: Regression odd-one-out accuracy for the embedding and logits layer for all supervised models trained on ImageNet-1K or ImageNet-21K. Dashed line indicates $x = y$.

## G.2    Can human concepts be recovered via linear regression?

In addition to linear probing, we performed $\ell_2$-regularized linear regression to examine models' ability to predict VICE dimension. This analysis helped us to further understand whether human concepts can be recovered from a neural network's representation space. Here, for each of the 45

representation dimensions, $j$, from VICE, we minimized the following least-squares objective

$$\underset{\boldsymbol{A}_{j,:},b_j}{\arg\min} \sum_{i=1}^{m} (Y_{i,j} - (\boldsymbol{A}_{j,:}\boldsymbol{x}_i + b_j))^2 + \alpha_j \|\boldsymbol{A}_{j,:}\|_2^2, \tag{3}$$

where $Y_{i,j}$ is the value of the $j^{\text{th}}$ VICE dimension for image $i$, $\boldsymbol{x}_i$ is the neural network representation of image $i$, and $\alpha_j > 0$ is a regularization hyperparameter. Each dimension was optimized separately with $\alpha_j$ selected via CV using grid search (details are in Appendix B.3).

The results from this analysis corroborate the findings from § 3.2: models trained on image/text data and ViT-G/14 JFT consistently provided the best fit for VICE dimensions, while AlexNet and EfficientNets showed the poorest regression performance. Furthermore, we investigated whether the recovered VICE dimensions show better alignment than the original network embeddings. All models were evaluated on the THINGS triplet task using a similarity matrix $\boldsymbol{S}$ with $S_{ij} := (\boldsymbol{A}\boldsymbol{x}_i + \boldsymbol{b})^T(\boldsymbol{A}\boldsymbol{x}_j + \boldsymbol{b})$, where $\boldsymbol{A}$ and $\boldsymbol{b}$ are obtained by stacking the optimizers from Equation 3, so $\boldsymbol{A}\boldsymbol{x} + \boldsymbol{b}$ is a linear regression from a neural network representation to the VICE representa-

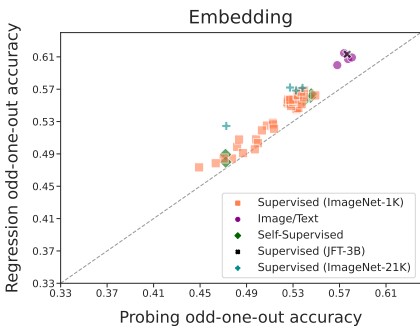

Figure G.2: Regression as a function of probing odd-one-out accuracies for all models in Table C.1

tion. In Figure G.2, we compare odd-one-out accuracies after linear probing and regression respectively. The two performance measures are highly correlated for both the embedding ($r = 0.960$) and logits ($r = 0.966$) layers. Note that odd-one-out accuracies are slightly higher for regression. We hypothesize that this is due to VICE being trained on all objects in the data so the transformation matrix learned in linear regression indirectly has access to all objects opposed to the transformation matrix learned during probing. Moreover, 2/3 of the objects were used for training the linear probe, whereas 4/5 of the objects were used to fit linear regression.

Figure G.3 shows the $R^2$ score for fitting a the same subset of VICE dimensions used in Figure 5 from embedding-layer representations.

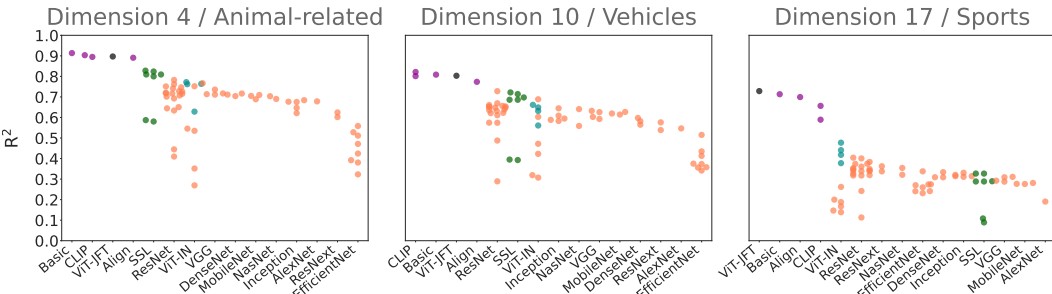

Figure G.3: $R^2$ scores for all models in Table C.1 after fitting an $\ell_2$-regularized linear regression to predict individual VICE dimensions from the embedding-layer representation of the images in THINGS. Color-coding was determined by training data/objective. Violet: Image/Text. Green: Self-supervised. Orange: Supervised (ImageNet-1K). Cyan: Supervised (ImageNet-21K). **Black**: Supervised (JFT-3B).

