# OpenReview forum: "Human alignment of neural network representations"
_NeurIPS.cc/2022/Workshop/SVRHM — SVRHM Poster_

### Official Review · Reviewer_MP4x · 2022-10-14
**Comprehensive test of human-CNN alignment on interesting new dataset**

**Rating:** 8
**Confidence:** 4

**Review:**

Pros:
- The paper is of high quality, describing an extensive set of experiments to conduct a comprehensive test of visual concept representations between human and CNNs, with clear writing and interesting results.
- Tackling the question of human-CNN alignment using the odd-one-out triplet task at the large scale of the THINGS database is I think original and novel.
- I believe the results to be significant as they point out that simply scaling networks will not automatically improve alignment with humans, and stress the importance of linear probing to reveal interpretable patterns of differences between the different classes of models in terms of their ability to predict human judgments.

Cons:
- Given the brevity of the format, this necessarily means a lot of methodology is going in the appendix, but this means that some results are only very briefly described in the main paper: e.g. the regression results are mentioned but not shown in the main paper, but are included in the Conclusion. The ‘royal objects’ class (?) is mentioned but not shown anywhere in the paper (main or Appendix).
- The proposed impact of the work for ML is not entirely clear: the authors suggest that the results should have ‘strong implications’ for the design of future algorithms, but how is that related to their results? What is are the implications for work focusing on transfer learning? Should we be training networks on (human) odd-one-out judgments perhaps?

Suggestions for improvement:

Abstract: I found the sentence on line 10 (“Using a sparse Bayesian model…”) confusing as it describes an analysis on the human data but draws a conclusion about neural network representations.

Motivating the work: The introduction states that comparing CNNs and humans on an odd-one-out task is measuring the alignment between them ‘more directly’ than comparing error consistencies or predicting brain measurements. I don’t really see why odd-one-out judgments are measuring similarity ‘more direct’ than errors; isn’t it just another interesting (but similarly ‘direct’) measure of internal representations?

Motivating the models: It would be good, in the limited space available, to provide some more context of the chosen models: why look at self-supervised models for example?

Under Appendix A (related work describing comparison of human and machine similarity judgments), it would be nice to cite Greene et al., (2016, JEP:G) who showed that ImageNet trained CNN does not capture human similarity judgments related to actions (“functions”), which resonates with the low alignment reported for the sports-associated dimension here.

Appendix E could benefit from more text motivating the figures: Figure E.1, right; this shows the same result as in Figure 2, right; is it just for reference? And what is the purpose of Figure E.2? According to the text is shows ImageNet accuracy but the axes labels say odd-one-out accuracy.

Similarly, I didn’t really see the benefit of the regression analysis reported in Appendix G, perhaps because it resulted in similar outcomes as the main probing analyses -- or perhaps because it was not clearly explained why one might expect different results for predicting VICE dimensions using regression versus using the linear probing done in section 3.3.

---

### Official Review · Reviewer_uvTC · 2022-10-16
**A thorough comparison between machine representations and human odd-one-out judgments.**

**Rating:** 9
**Confidence:** 4

**Review:**

# Summary
* Used human behavioral data from an odd-one-out similarity task as a window into human representations.
* Assessed the correspondence between numerous model representations and human behavior/representations.
    * Used two metrics: *zero-shot odd-one-out accuracy*, *probing odd-one-out accuracy*
* Inferred the feature dimensions with the least/most agreement between human and machine representations.
* General conclusions:
    * Model scale and architecture minimally influences alignment with behavior.
    * Training data and objective function influences alignment with behavior. Specifically, using image and text data boosted model alignment.
    * Representations from earlier layers perform worse than higher layers.

# Pros

* The different metrics and analysis provided a coherent and layered analysis: going from simple (*zero-shot odd-one-out accuracy*), to flexible (*probing odd-one-out accuracy*), to targeted and interpretable (linear regression with VICE dimensions). The linear regression with VICE dimensions was particularly novel and original.
* Used a large and diverse set of models.
* Assessed models at multiple levels: logits, penultimate, and lower representations.
* Organization is clear (with helpful section/subsection titles) and prose is polished.
* The key results are clearly stated throughout.
* The significance of the work is clear and plausible:
    * There is still a substantial gap between machine and human representations.
    * Training on the multiple types of data and carefully choosing the objective function seems a more fruitful path forward versus using larger and more complex architectures.

# Cons

### Major
* No major concerns.

### Minor
* The authors seem to have overlooked Roads and Love (2021) that used a similar behavioral task and analogous metric to the authors' "zero-shot odd-one-out accuracy". They reached similar---although less exhaustively supported---conclusions about the disconnect between machine and human representations (such as increasing architecture complexity does not improve alignment with human behavior).
* The analysis using VICE dimensions (Section 3.3) was not as clear as the other sections. The details in the Appendix helped, but the main text would benefit from more hand-holding. For example, I was disoriented when the Conclusion stated "...human alignment is weak for concepts that depict sports-related or royal objects, for which the interplay between a minimal bias for shape or texture and strong semantic knowledge appears to be crucial (lines 196-198)." This is incredibly interesting, but some logical steps are missing.
* Different models can have a preferred kernel for computing similarity between its representations. For example, self-supervised or contrastive models are often trained using a particular kernel. Does the choice of cosine similarity (in the zero-shot metric) disadvantage models that were trained using a different kernel?

## References
* Roads, B. D., & Love, B. C. (2021). Enriching ImageNet with human similarity judgments and psychological embeddings. In Proceedings of the IEEE/CVF Conference on Computer Vision and Pattern Recognition (pp. 3547-3557). DOI: 10.1109/CVPR46437.2021.00355.

---

### Official Review · Reviewer_DPNh · 2022-10-17
**Some concept representations are human-aligned, diverse training datasets result in better alignment**

**Rating:** 6
**Confidence:** 3

**Review:**

The paper studies how well human representations of concepts align with representations of neural network responses for models with different objective tasks and architectures, trained on different datasets. Behavioral responses on odd-one-out task are taken as a proxy for human concept representation. The study finds some concepts are more well-represented in networks than others. They observe no effect of model scale or architecture on the alignment, but training dataset and objective seem to have more impact. They use linear regression to predict VICE dimensions and find some concepts can be better predicted than others.

Pros:
1.  The authors probe an interesting question: whether the SOTA networks learn human aligned representations or not and what network choices affect them the most

2.  They consider a vast variety of models with various architectures, objectives, and datasets.

3. This work could be interesting for designing interpretable models, one could imagine testing whether a model is human aligned before it's implementation.

Cons:

1. For images in Fig 1, it is not mentioned which image neural nets choose as odd-one-out, it would be better to add that in.

2. Only 3 VICE dimensions are shown, the authors may find it worthwhile to add more concept dimensions to make their case stronger

3. The authors mention royal objects in the introduction and conclusion, but do not elaborate on them or show any examples of them, these should be included if they are mentioned in the paper.

---

### Official Review · Reviewer_ukxa · 2022-10-17
**A useful framework for studying model-human alignment; results hint toward meaningful takeaways; need for tighter controls**

**Rating:** 7
**Confidence:** 3

**Review:**

This paper attempts to study how different inductive biases (model architecture, training dataset, learning algorithm) impact the alignment between model and human similarity judgments after training. The authors evaluate a set of several dozen publicly-available pretrained DNNs, including supervised and self-supervised ConvNets, Vision Transformers, and language-aligned models. As a target for the models, the authors use behavioral data from Hebart et al. 2020, which maps the major conceptual dimensions underlying human object similarity judgments at scale via an odd-one-out triplet task, where subjects identify the member of an image triplet that is most dissimilar from the other two images. This large-scale, reliable dataset represents a rich and useful target for assessing model-human alignment.

The authors use two different mapping methods to assess correspondence between model and human behavioral judgments: "zero-shot" accuracy, where model triplet behavior is assessed via cosine similarity between output representations; and, "probing" accuracy, where a linear mapping is fit to the model representations in an attempt to maximize the potential match to human behavioral responses.

In computing the behavioral performance of the models, the authors find that:

- ImageNet top-1 accuracy very weakly explains "zero-shot" model-human alignment (this motivates a systematic analysis of the specific inductive biases that impact human alignment)
- Holding architecture (ResNet50) and training set (ImageNet) constant, the use of different data augmentation schemes or objective functions yield different levels of human alignment (ranging from accuracy around 0.4 to around 0.53).
- Holding training set and objective function constant and varying architecture, ImageNet performance varies but human alignment levels do not, suggesting that architectural variability is not a major inductive bias impacting model-brain behavioral alignment
- Different self-supervised training schemes appear to yield different levels of alignment (MoCoV2 best, Jigsaw Puzzle worst)
- There is no relationship between model width/depth and odd-one-out accuracy
- Models trained on image/text data, or more diverse datasets, achieve the highest degree of alignment (better than ImageNet-trained models).
- Assessing alignment via the linear probe improves accuracy levels across the board, and reduces the variance in alignment levels across models. doesn't really change the authors' conclusions.

The authors then attempt to study whether specific human concepts show better or worse model-human alignment for 3 specific concepts (animal-related, vehicles, and sports). They find that some major concepts (e.g. food and animals) can be easily recovered from network representations, while others are not.

At a high level, the pattern of results suggests that models trained on larger and more diverse datasets do the best job explaining human behavioral judgments, but that none of the inductive biases under investigation alone are sufficient to train models with extremely human-like conceptual representations.

Pros

- The paradigm of leveraging high-quality, large-scale human neural and behavioral datasets to assess alignment is a promising avenue forward toward understanding the shortcomings of current models and moving toward more powerful explanatory models of human perception and cognition.
- The authors recognize the importance of attempting to isolate specific inductive biases (e.g. model depth) while holding others constant in order to try and provide generalizable insight into how different constraints on representation learning affect alignment levels.
- The authors test a reasonably wide swath of models - enough so that the main findings are believable, even without conducting statistical tests on the effect sizes
- The use of both unweighted (zero-shot) and weighted (linear probe) mapping methods is an important step that helps contextualize the findings and adds robustness
- The writing is generally clear and concise
- The main text introduction does a good job situating this work within its historical context and poses the key research question in a clear way
- The analysis of whether human-alignment is concept-specific is a valuable addition to the paper that helps to unpack the main effects observed in the model-human comparisons.

Suggestions

- Analyses that seek to isolate the impact of a specific inductive bias on alignment call for very careful control of the other factors that might vary between models. For example, it is hard to interpret the lack of an effect of # parameters on alignment levels if models are also varying in their training diet (e.g. ImageNet 1k vs. 21k). A stronger version of the main analyses conducted here would involve a more systematic effort to control all external factors of variation and a large enough set of models that statistical tests could be performed to confirm that a specific inductive bias has a distinct impact on alignment levels under a particular mapping scheme. Another example is that language-aligned models such as CLIP cannot be directly compared to models trained on ImageNet-1K to assess the impact of language alignment, since the CLIP training set is orders of magnitude larger and it is unclear whether improvements would be due to scaling of the training set or the language alignment constraint. The interpretability of the paper would be improved if the authors were more explicit about scenarios where multiple factors of variation between models may impact their relative differences in human alignment.
- It seems that a main takeaway is that models trained on massive amounts of data (JFT-3B) perform better? Given this hint in the data, it would be nice if the authors included an analysis that systematically varied the number and/or diversity of training images and measured the impact on alignment.
- The concept-specificity analyses could be more systematic - what is going on with all the other dimensions, other than animal-related, vehicles, and sports?
- In the main text, there is no real justification for the analyses involving CIFAR-100 - I was confused about the rationale
- Figure 3, top right: it is unclear which objective function is being used and held constant to assess the impact of architecture - this could be made more explicit. It also begs the question whether the relative unimportance of architecture holds for different objective functions
- Figure 3, top left: it was a bit confusing that objective functions and data augmentation schemes were grouped together in this analysis. These feel like two distinct categories of inductive bias, and perhaps should be examined separately.
- In describing the interesting result that different self-supervised learning schemes yield different levels of alignment, the authors could do more to speculate about why e.g. MoCoV2 performs best among the Siamese methods – are there general things we can conclude from this?
- It might be obvious that chance performance levels are 33.3%, but it would be helpful for the sake of interpretation if y-axes were held constant across figures 2-4 and if chance performance were labeled. It would be easier to contextualize both the relative performance of different models and the absolute degree of alignment being achieved.
- In section 3.2, it seems strange to call SWAV an outlier given how many other models are close by - maybe a different wording would avoid confusion about whether the listed models are actually statistical outliers

Overall, I feel that this is a solid paper that merits acceptance to SVRHM. Addressing the above suggestions would strengthen the paper and aid interpretability of the main findings.